# Phosphorus-Containing Flame Retardants from Biobased Chemicals and Their Application in Polyesters and Epoxy Resins

**DOI:** 10.3390/molecules24203746

**Published:** 2019-10-17

**Authors:** Jacob Sag, Daniela Goedderz, Philipp Kukla, Lara Greiner, Frank Schönberger, Manfred Döring

**Affiliations:** 1Fraunhofer Institute for Structural Durability and System Reliability LBF, D-64289 Darmstadt, Germany; jacob.sag@lbf.fraunhofer.de (J.S.); daniela.goedderz@lbf.fraunhofer.de (D.G.); philipp.kukla@lbf.fraunhofer.de (P.K.); lara.greiner@lbf.fraunhofer.de (L.G.); frank.schoenberger@lbf.fraunhofer.de (F.S.); 2Ernst-Berl Institute for Chemical Engineering and Macromolecular Science, Technische Universität Darmstadt, D-64287 Darmstadt, Germany

**Keywords:** biobased polymers, polyester, epoxy resin, flame retardant, phosphorus-containing flame retardants

## Abstract

Phosphorus-containing flame retardants synthesized from renewable resources have had a lot of impact in recent years. This article outlines the synthesis, characterization and evaluation of these compounds in polyesters and epoxy resins. The different approaches used in producing biobased flame retardant polyesters and epoxy resins are reported. While for the polyesters biomass derived compounds usually are phosphorylated and melt blended with the polymer, biobased flame retardants for epoxy resins are directly incorporated into the polymer structure by a using a phosphorylated biobased monomer or curing agent. Evaluating the efficiency of the flame retardant composites is done by discussing results obtained from UL94 vertical burning, limiting oxygen index (LOI) and cone calorimetry tests. The review ends with an outlook on future development trends of biobased flame retardant systems for polyesters and epoxy resins.

## 1. Introduction

Regarding the definition of the term “biobased” by IUPAC, a biobased material is comprised partly or wholly of biological products generated from biomass. Biobased materials must not necessarily be biodegradable, biocompatible or environmentally friendly, particularly when they are used to replace their counterpart polymer derived from petrochemical resources [1]. In terms of research and development, a growing interest has emerged of new polymers derived from biobased chemicals or biomass. There is also an increasing interest in a greener synthesis process of flame retardants using sustainable substances or plant-derived chemicals, particularly when considering that petroleum resources are finite. This would also lower the environmental impact. Plant proteins, mainly derived from wheat, soybean and corn or plant oils like soybean, corn and flax oil or plant starches like carbohydrate polymers and cellulose are plant-derived products from renewable sources that can be used neat or modified for the replacement of petrochemicals in certain applications.

Biobased polymers or flame retardants can be made partially or totally from renewable sources. The biobased content in materials is determined by using the standard ASTM D6866-11 test [2] by radiocarbon analysis which quantifies the total organic carbon content in the product. For the utilization of substances derived from renewable resources it is imperative to fulfill the following requirements: sustainability, affordability, compatibility and durability [3,4,5].

Nowadays, there are many processes available for the production of biobased chemicals in biotechnical processes involving the extraction and modification of biomass and fermentation of natural products like sugars or plant material. The development of biobased flame retardant formulations as a whole includes biobased flame retardants and biobased polymer matrices. Engineering plastics like poly(ethylene terephthalate) (PET), poly(butylene terephthalate) (PBT), etc. and biobased polymers like (poly(lactic acid) (PLA) can be produced by using biobased chemicals. Products made from biobased materials are not necessarily biodegradable, particularly when they are chemically modified. Most polymers are known to withstand degradation which is a desirable property, in particular when they are exposed to weathering such as in the case of wind turbine rotor blades, etc. [6,7], but this resistance to degradation is a challenge for biodegradability. Müller et al. reported a depolymerization of PET by using a hydrolase isolated from *Thermobifida fusca* resulting in water soluble oligomers/monomers [8]. The biodegradability has to be considered in the development of materials produced with biobased materials to generate sustainable materials. Biodegradability and synthesis pathways with biobased materials are still a big challenge in the future, because due to the plant growth there has to be a balance between the application of plant material for food and raw materials for greener solutions. The land use for biobased polymers (PLA, PHA, PTT, PBAT, starch blends, etc.) in 2017 was 672,000 ha which corresponds to 0.005%. The global agricultural area requires 5 billion ha (3.65%). An increase up to 1,038,000 ha (0.008%) is predicted by IfBB for biobased polymers in 2022 [3,4,5,9].

Biobased polymers include all polymers which are made of renewable biosources resulting in biodegradable or non-biodegradable biobased polymers. The biobased non-biodegradable part of biopolymers still represent a greater part of the biobased polymers production capacity in 2017 and there will be a slight change until 2022 (Figure 1). The main part of biobased non-biodegradable biobased polymers is represented by bio-PET derived from plant material. The Coca-Cola Company developed the PlantBottle, which is a blend of plant-derived material (up to 30%) and petrochemical material in 2009 [10,11]. PLA use as a biodegradable polyester will rise from 10.5% in 2017 to 18.7% in 2022 [9].

Table 1 summarizes selected chemical building blocks which can be produced by a biotechnical process for certain applications [12,13]. Combustible plastics which are near electrical or heat sources require flame retardant solutions. Due to the development of electric mobility in the last years and limited petrochemical resources biobased polymers will be one of the target materials for the future in this and other segments. Biobased polymers and biobased flame retardants have a huge potential for greener polymer synthesis and flame retardant solutions and contribute to a reduced environmental impact in this context. This review covers biobased phosphorus-containing flame retardants for epoxy resins and polyesters. The application range of polyesters includes synthetic fibers [14] in home furnishing or car upholstery and foams in construction materials like wind turbine blades [15]. Their resistance to stretching, most chemicals and wrinkling are characteristics of polyesters which make them very interesting for the textile industry [16]. Polyesters, which are made of renewable raw materials and are biodegradable, represent a greener solution in terms of sustainability than biopolyesters which are not biodegradable. Certain polyesters possess biodegradability from the outset like PLA or PBS and can be produced with renewable resources. The consideration of the life cycle assessments (LCA) and the comparison between petroleum based polyesters and biobased polyesters also includes the evaluation of the environmental profile regarding emissions during the production process or the agriculture operations of the raw material extraction. As long as the comparison of the life cycle assessment is not favorable regarding environmental aspects, a biobased polymer implicates no superiority to its petrochemical based derivative. Chen et al. worked out the life cycle assessment of petroleum and biobased PET bottles [17].

Epoxy resin systems form thermosets by crosslinking of multifunctional epoxy monomers and curing agents. Since their properties, like mechanical properties, chemical resistance, low cost, relative low curing shrinkage [18,19] are exterior to other materials, they are used in many different applications like coatings, composites or high performance materials [20,21]. The most common epoxy resin is bisphenol A diglycidyl ether (DGEBA), which is synthesized from bisphenol A (BPA) and epichlorohydrin [22]. For epichlorohydrin, a synthetic route from glycerin as a renewable source is economically viable [23], but there is no similar route for BPA [24]. Another disadvantage of DGEBA-based epoxy resins is their flammability [25]. Therefore, there is put much effort in finding new flame retardant agents for epoxy resins based on green chemicals. There are reactive flame retardants, that are incorporated into the network by preformulation reaction or throughout the curing process, and non-reactive flame retardants. Reactive flame retardants with two or more reactive groups operate as a hardener or as a resin [26,27] as an additional feature. Non-reactive flame retardants can be multifunctional improving mechanical, thermic, electric and other properties [28,29,30,31].

## 2. Synthesis of Phosphorus-Containing Biobased Flame Retardants

There are different biobased materials which possess no suitable flame retardant efficiency by themselves, but a chemical modification can lead to a potential flame retardant agent. For example, biobased alcohols like isosorbide, pentaerythritol, vanillin, cardanol, eugenol, glycerol, etc. or double bond-containing structures like itaconic acid (IA) are starting materials which can be obtained from biomass. This chapter summarizes different phosphorylation methods for the synthesis of phosphorus- containing flame retardants used for polyesters and epoxy resins. The major part of the phosphorylating agents is not biobased, but the final flame retardant product has an increased biobased ratio due to the phosphorylation of biobased starting materials.

### 2.1. Biobased Flame Retardants for Poly (Lactic Acid)

One of the most used biobased chemicals for the synthesis of “green” flame retardants is pentaerythritol (PER, Scheme 1). It can be synthesized by conversion of acetaldehyde with formaldehyde, which both can be produced from biomass-to-liquid (BtL) process or carbon monoxide hydrogenation [79]. PER is often used in intumescent flame retardant systems (IFR) as a carbonization agent along an acid source and a blowing agent [80,81,82]. For the best synergistic effect between the components which are required for intumescence, various research groups combined them into one molecule. Tao et al. synthesized a phosphazene cyclomatrix network polymer poly(cyclotriphosphazene-co-pentaerythritol) (PCPP I, Scheme 1) using PER and hexachlorocyclo-triphosphazine (HCCP). A high char residue (68 wt%) at 600 °C after thermogravimetric analysis indicates that PCPP is an efficient char forming agent [83].

Zhan et al. [34] reported the synthesis of spirocyclic pentaerythritol bisphosphorate diphosporyl melamine (SPDPM) from spirocyclic pentaerythritol (SPDPC, Scheme 1) and melamine. After decomposition, SPDPM possesses high char residues (40 wt%) at 600 °C [84]. Xuan et al. synthesized the PER derived caged bicyclic pentaerythritol phosphate alcohol (PEPA, Scheme 1) for novel intumescent flame retardants, IFR-I and IFR-II (Scheme 1). Similar to other systems, TGA results of IFR-I showed a high residue percentage of 45 wt% at 600 °C [85]. Jing et al. described the synthesis of another PEPA based flame retardant using diphenolic acid, which can also be derived from biomass [86,87]. TGA data of the novel biobased polyphosphonate (BPPT, Scheme 1) showed a residue weight of 41 wt% at 600 °C.

Further flame retardants derived from biobased chemicals were synthesized by Lin et al. and Zhao et al. using 1,2-propanediol and vanillin (Scheme 2a,b). The TGA curves of both compounds indicate a flame retardant action mainly in the gas phase [88,89]. Wang et al. synthesized an inherently flame retarded PLA (PPLA I, Scheme 2c) via-chain extending reaction of dihydroxyl terminated pre-poly(lactic) acid (Pre-PLA, Scheme 2c) with ethyl phosphordichloridate and melt blended it with PLA [90]. Another intrinsically phosphorus-containing PLA (PPLA II, Scheme 2d) was synthesized by Yu et al. using L-lactide, P-DDS-Ph and hexamethylene diisocyanate (HDI) (Scheme 2d) [91].

### 2.2. Biobased Flame Retardants for Poly (Ethylene Therephthalate) and Poly (Butylene Succinate)

Besides isomannide and isoidide, isosorbide as 1,4,3,6-dianhydrohexitol can be obtained from starch via glucose as intermediate. The synthesis of isosorbide from starch requires an enzymatic process degrading the starch into D-glucose and D-mannose. D-Glucose is hydrogenated into D-sorbitol which can be dehydrated to isosorbide (Scheme 3). In a similar way, isomannide is obtained from D-mannose via D-mannitol [92,93,94].

The phosphorylation towards *bis*-phosphorus esters can be done by a nucleophilic substitution with the phosphorus chlorides or by Atherton-Todd reaction with 9,10-dihydro-9-oxa-10-phospha-phenanthrene-10-oxide (DOPO, Scheme 4) [95]. Different phosphorus modified isosorbide-containing flame retardants were analyzed in PLA [96] and PBS [97].

Hu et al. used phenylphosphoric dichloride, dichlorophenylphosphate, diphenylphosphate chloride, diphenylphosphoric chloride and DOPO as phosphorous compounds resulting in different chemical environments of the phosphorous atom in the flame retardant. A mass loss of 2% can be observed in the range of 289–321 °C in nitrogen atmosphere for all phosphorus-containing isosorbide flame retardants [97,98].

Lignin is a phenol-containing cross-linked biomacromolecule that can be isolated by two different extraction procedures (alkali and organosolv). The flame retardant efficiency of neat lignin is already known for different polymer systems [51,52,99]. Ferry et al. compared the fire behavior of lignin obtained from different extraction procedures in PBS. The two lignin types differentiate in the decomposition temperature; organosolv lignin has a 30 °C higher decomposition temperature in the first decomposition step (349 °C). Three main decomposition steps are taking place during the decomposition of lignin. In the first step (230–260 °C), the release of low molecular weight products of a propanoid side chain cleavage takes place. The second step is the main degradation step (275–450 °C) where large amounts of methane are released. The formation of char and the release of dihydrogen take place above 500 °C. Ferry et al. compared the two neat lignins also by using cone calorimetry. By using a grafting onto process, lignin can be modified with phosphorous compounds and the flame retardant efficiency can be enhanced [53].

Itaconic acid is a biobased monomer which can be used for the production of aliphatic esters containing DOPO and can be gained by the fermentation of sugars. There is a growing market of itaconic acid due to the versatility of this unsaturated trifunctional diacid. It can be converted to polyamides with diamines, polyesters with diols, polyitaconic acid or polyacrylates with methacrylic acid derivatives [100]. Pospiech et al. developed different linear aliphatic polyesters with biobased contents up to 100% containing itaconic acid. An increasing DOPO monomer content leads to an increasing glass transition temperature. The obtained polyesters (Scheme 5) were compared to PET, PTT and PBT by using pyrolysis combustion flow calorimeter (PCFC), also called microscale combustion calorimeter (MCC) [101,102].

Glycerol is a trifunctional alcohol which can be modified in a two-step process resulting in a star shaped highly thermostable DOPO-containing glycerol triacrylate (GL-3DOPO, Scheme 6) by using acryloyl chloride. The decomposition of GL-3DOPO starts at 360 °C in nitrogen atmosphere and at 368 °C in air atmosphere. Xie et al. melt blended GL-3DOPO with engineering plastics like PET, PBT, PA6, PA66 and PC. GL-3DOPO decreased the melting temperature and the glass transition temperature of the melt blended polymers [103].

Phosphorus chlorides or P-H-compounds can be used for phosphorylations within the context of Atherton-Todd reactions, whereas the latter requires carbon tetrachloride as chlorinating agent. In terms of green chemistry, carbon tetrachloride can be replaced by *N*-chlorsuccinimide resulting in succinimide as byproduct. By using *N*-chlorsuccinimide as chlorinating agent, flame retardant synthesis of DOPO-PEPA (Scheme 7) is more sustainable than using carbon tetrachloride. DOPO-PEPA is used as flame retardant in PET [104,105], PBT [104] and epoxy resins [104,106].

Deoxyribonucleic acid (DNA, Figure 2) is a biomacromolecule and provides a source of acid by the phosphate group. The deoxyribose moiety has high carbon content and the nitrogen containing bases have an effect as blowing agent. In sum DNA is a good basis for an intumescent flame retardant system. DNA revealed as a suitable flame retardant for surface treated cotton fabrics, PET foams and bulk samples of different polymers [105,107,108,109,110,111]. A DNA containing coating provides a physical barrier through the formation of a carbon-rich protective shield when exposed to a heat source [112].

Milk contains casein as phosphorus-based proteins which are mainly used as food ingredient. Caseins possess high phosphorus content due to seryl-phosphate groups or phosphoserine groups depending on the components of the casein. The fire behavior of cotton, polyester or fabrics containing blends of both with deposited casein was investigated by Alongi and Carosio et al. Casein promotes the polyester cyclization and the char formation resulting in a protective layer [112,113,114].

### 2.3. Biobased Flame Retardants for Epoxy Resin Systems

#### 2.3.1. Reactive Flame Retardant Epoxy Monomers

Animal testing has revealed that BPA has negative effects on reproduction and neurobehavioral effects, a potential to favor cancer and metabolic disorders in small dosages as well as other negative effects at high dosages [115,116,117]. Many research groups have tried to find a new, renewable raw material for epoxy resins [118], but also intrinsic flame retardant, biobased epoxy resins are developed based on eugenol [40,119], daidzein [120] components of soybeans [121], furan derivatives [40], itaconic acid [45,122] or pentaerythritol [123]. Another approach is incorporating flame-retardant silicon- [124] or phosphorus- [45] containing groups into the backbone as substituents in epoxy resins or curing agents [48]. Flame retardant epoxy resins are typically synthesized by:incorporation of phosphorus-containing groups into an epoxide component orincorporation of phosphorus-containing groups into a molecule with a following epoxidation of unsaturated side chains.

Ménard et al. [62] introduced epichlorohydrin to the biobased phloroglucinol (benzene-1,3,5-triol) developing a trifunctional molecule, shown in Scheme 8. The zinc chloride catalyzed oxirane ring opening reaction with triethyl phosphate leads to a flame retardant monomer (P2EP1P, Scheme 8).

Ma et al. [45] used a catalyzed reaction to add 9,10-dihydro-9-oxa-10-phosphaphenanthren-10-oxide (DOPO) to the double bond of itaconic acid. They determined two synthetic routes to obtain epoxides, shown in Scheme 9. The direct epoxidation with epibromohydrin (Scheme 9b) leads to the same product (EADI) with slight lower yield than the esterification with allyl bromide followed by an oxidation with a peroxide like *meta*-chloroperbenzoic acid (Scheme 9a).

Lligadas et al. [60,125] esterified a phosphorus containing molecule (DOPO-HQ) with biobased 10-undecenoyl chloride followed by epoxidation the terminal double bond, shown in Scheme 10. This leads to the phosphorus-containing flame retardant DOPO-III. The starting compound DOPO-HQ is synthesized by the addition of DOPO to *p*-benzoquinone.

Caillol et al. [41] analyzed different biobased starting materials for flame retardant epoxides. Cardanol or eugenol and phosphoryl chlorides undergo a condensation reaction with HCl as condensate, shown in Scheme 11. Eugenol is a phenylpropanoid, which can be obtained from clove oil, but different plants like sweet basil produce eugenol as well [126]. Depending on the quantity of functional groups in the phosphorus-containing derivative, two or three eugenol molecules are linked leading to two or three epoxide functional groups after epoxidation. The reaction pathway and the products TEEP, DEEP and DEEP-Ph are shown in Scheme 11.

Cardanol, on the other hand, being non-edible and a by-product of the cashew industry, is a promising source for phenolic oils [127]. Consequently, different approaches to use cardanol have been examined, like the usage as fuel [128], process chemical [129,130] or raw material for special chemicals [131,132]. Cardanol does not have a defined number of double bonds, it can vary between 0 and 3. Therefore, the number of epoxy groups can vary with the same reaction route. Furthermore, the epoxidation agent hydrogen peroxide reacts partially with terminal double bonds [41]. The reaction of cardanol and phosphoryl trichloride followed by epoxidation led to the flame retardant TECP shown in Scheme 12.

#### 2.3.2. Reactive Flame Retardant Curing Agents

Not only epoxy resins, but also curing agents can be modified. As there is a broad range of different phosphorus-containing hardeners (e.g., by Toldy et al. [133]), we will focus on biobased hardeners containing DOPO.

In general, DOPO reacts with benzaldehyde in a Pudovik-type reaction by electrophilic addition [134]. After hydrolysis of the secondary hydroxyl group, a nucleophile can react with the transient carbocation via nucleophilic addition and deprotonation. Dai et al. [135] combined the DOPO-adduct of 4-hydroxybenzaldehyde and aniline to incorporate nitrogen to the flame retardant (DOPO-PHM, Scheme 13). Lin et al. [136] compared the reactivity of the DOPO-adducts of vanillin and acetovanillone under alkaline condition adding phenol after hydrolysis. This reaction path is not suitable for vanillin since the secondary carbocation is not stabilized for further reactions. The resulting bisphenolic structure with acetovanillone (hydroxyl PES, Scheme 13) can be modified by ring closing reactions with furfurylamine and paraformaldehyde. This reaction and the flame retardant benzoxazine are shown in Scheme 14. These benzoxazines have up to six different possible reactive sites [137]. Liu et al. [36] added DOPO to terephthaldicarboxaldehyde and the adduct TDCA-DOPO was polymerized with phenol to a novolac compound (Ar-DOPO-N, Scheme 13), that is suitable as flame retardant in novolac epoxy resins. This reaction path was also used by Wang et al. [138] for their synthesis of TDCA-DOPO. Then it was condensed with SPDPC leading to PFR (Scheme 15).

Yang et al. [57] investigated a different approach, shown in Scheme 16. They examined myrcene, an essential oil which is already used for different syntheses like the synthesis of menthol or geraniol [59]. By a Diels-Alder reaction, myrcene reacts with maleic anhydride leading to the structure of an anhydride hardener [139]. In the next step DOPO is introduced into the molecule by addition reaction. The resulting compound MMDOPO is shown in Scheme 16.

To obtain a flame retardant pressure-sensitive adhesive material (PSA), Wang et al. [69] modified DOPO-HQ with ethylene spacers [140] and used a polycondensation reaction with a prepolymeric ester from sebacic acid, yielding a polymeric hardener shown in Scheme 17.

#### 2.3.3. Non-Reactive Flame Retardants

Multifunctionality is not limited to flame retardant hardeners or epoxy resins. For flame retardants, several different functionalities are reported, like anti-aging [141], synergistic [142], nano-filler [143], toughening [18] or fiber protection [21] effects.

The reactivity of oxirane groups can be used to add different functional groups. For example, Hu et al. [18] combined TECP (the phosphate of cardanol after epoxidation) and DOPO, shown in Scheme 18. The resulting non-reactive flame retardant acts additionally as a toughening agent in epoxy resins.

Wang et al. [143] synthesized nano-filler flame retardants from phytic acid by a hydrothermal method. Phytic acid (see Scheme 19) is a phosphorus reservoir that acts as antioxidant for plants like grains or beans [144]. It was converted with melamine to the curing agent PAMA. Another approach are inclusion complexes that show better flame retardant efficiency. Zhao et al. [145] combined cyclodextrine, a degradation product of starch, as charring agent and *N*,*N*′-diamyl-*p*-phenylphosphonicdiamide (P-MA, synthesized from amilamine see Scheme 20) as an acid source [146]. 

There are other promising biobased systems, which need to be tested for flame retardancy. Fache et al. synthesized epoxy monomers from lignin [54,147]. Different research groups have developed biobased flame retardants without determining other applications. Jin et al. added DOPO twice to acrolein achieving V-0 at 3 wt% flame retardant content in DGEBA/DDM (Scheme 21a) [148]. Acrolein is available from biobased alcohols via dehydration [149]. Via Atherton-Todd phosphorylation, Howell and Sun [150] combined diethyl tartrate and DOPO to DT-DOPO, shown in Scheme 21b. Tartrate precipitates as insoluble salts in wine production [74].

Daniel et al. investigated the fire behavior of di-DOPO-isosorbide (DDI), diphenylphosphate isosorbide (DPPI), diphenylphosphinate isosorbide (DPPII) and isosorbide bis-(diethylphosphate) (IDEA) (Scheme 4) in epoxy resins containing 1 wt% phosphorus. The TGA results show an increased residue with increasing oxygen content in the chemical environment of the phosphorous atom in nitrogen atmosphere. The phosphate IDEA has the lowest onset temperature with 152 °C. The thermal decomposition is initiated by the loss of the appropriate phosphorus acid, promoting highly crosslinked char due to Friedel-Crafts like reactions, dehydration and other reactions taking place in the degrading polymer.

## 3. Biobased Flame Retardants in Polyesters

### 3.1. Poly (Lactic Acid)

Due to its excellent properties PLA has reached commercial-scale production in recent years [151]. PLA features outstanding properties like biocompatibility, good mechanical properties, high transparency, low toxicity and flexible processability, thus it is suitable for a variety of applications [152,153,154]. Further utilization is limited by its poor fire performance, characterized especially by a high ignitability and flaming drips. Therefore flame retardants such as phosphorus-containing compounds, nitrogen-containing compounds and mineral fillers were incorporated and their flame retardant efficiency in PLA was investigated [155,156,157]. While there are already reviews covering the work of various research groups evaluating flame retardants in PLA, this review focuses solely on biobased flame retardants and their synergistic formulations [155,156,157].

Upon incorporation of only 5 wt% PCPP I (Scheme 1) PLA reaches a V-0 in the vertical UL94 burning test setup. Additionally, with increasing concentration of PCPP I (10 wt%) melt dripping was eliminated due to the formation of a cohesive char layer and the LOI value of PLA (21%) was increased to 26% while the peak heat release rate (pHRR) was decreased by 16% in the cone calorimeter test [83].

SPDPM (Scheme 1) was melt blended with PLA in various concentrations in the range of 5–25 wt%. The resulting PLA composites (5 wt%, and 15 wt%) passed the UL94 test with a V-2 rating. At a concentration of 25 wt% SPDPM, a V-0 rating was obtained and the LOI value was increased to 38% while also reducing the dripping behavior. In microscale combustion calorimetry (MCC) experiments the pHRR value was decreased by 39% from 475 W/g^−1^ to 291 W/g^−1^ [84].

The addition of IFR-I and IFR-II (Scheme 1) in PLA improved the flame retardancy of PLA/IFR composites passing the vertical UL94 test. While an incorporation of 30 wt% IFR-II only reached a V-1 rating an increase of the concentration up to 20 wt% IFR-1 lead to a V-0 classification. Moreover, the PLA/IFR-I (20 wt%) composite showed an LOI of 36% and a decreased heat release capacity (HRC) of 33% in pyrolysis combustion flow calorimetry (PCFC) [85].

4wt% BPPT (Scheme 1) in PLA resulted in a V-0 rating in the vertical UL94 burning test and an increased LOI of 33%. The PLA/BPPT blends show no significant difference to neat PLA in cone calorimeter test due to the lack of a char layer, which indicates a gas phase activity mainly responsible for the flame retardant effect [86]. In combination with polyethyleneimine-modified graphene oxide (M-GO), the proportion of flame retardant was further reduced to 3 wt% (2.4 wt% BPPT, 0.6% M-GO) while still passing the vertical UL94 test with a V-0 classification [158].

Upon addition of 5 wt% VP (Scheme 2a) in PLA the corresponding composites obtained a V-0 classification and the LOI improved to 26%. In the cone calorimeter test, the pHRR decreased by 10% and a less developed char layer along with a significant increase of evolved CO during the test indicate that gas phase action was the main flame retardant mechanism [88].

Melt blending of 7 wt% PCPP II (Scheme 2b) results in a V-0 rating with no melt dripping behavior in the UL94 test and an increased LOI value of 28% [89]. Further investigation of the thermal degradation behavior of the PLA/ PCPP II system using TGA, TGA-FTIR and Py-GC-MS revealed an improved thermal stability after 10% decomposition of the blend took place and confirmed the gas-phase action of PCPP II [159].

Upon incorporation of 5 wt% of PPLA I (Scheme 2c) PLA reached a UL94 V-0 rating and a LOI value of 34%. The pHRR was decreased by 23% from 436 kW/m^−^**²** to 337 kW m^−^² [90]. PPLA II (Scheme 2d) achieved a V-0 rating when 10 wt% were added in PLA. A LOI value of 28% and a decreased pHRR by 19% was obtained [91].

Isosorbide-based polyphosphonate (PPPI, Scheme 3) was added to PLA via melt blending and the corresponding composites were characterized by vertical UL94 test setup and cone calorimeter test. Even though the addition of 15 wt% PPPI did not significantly reduce the pHRR a V-0 classification was obtained [34].

Besides the incorporation of novel synthesized biobased flame retardants, various research groups used another approach to produce flame retardant PLA by using biomass derived charring agents with little or no chemical modification in flame retardant systems. A prominent example used in intumescent flame retardant formulations is lignin. Réti et al. evaluated the flame retardant efficiency of lignin (10 wt%) in combination with APP (30 wt%) and compared it to APP/PER (10/30 wt%) and APP/starch (10/30 wt%) formulations [160]. Although the highest LOI values were obtained with the PLA/APP/PER composite (60%) followed by the PLA/APP/starch (40%) and the PLA/APP/lignin (32%) composite, only the materials containing lignin or starch reach a V-0 classification in UL94 test. In the cone calorimeter test all formulations have a decreased pHRR of at least 40% up to 60% due to the formation of intumescent protective layers with the APP/PER formulation being the most efficient one. 

Zhang et al. compared the flame retardant effect of virgin lignin and urea modified lignin (UM-lignin) in combination with APP in PLA [99]. While the formulations containing APP/lignin (18.4 wt%/4.6 wt%) and APP/UM-lignin both showed similar LOI values (33–34%) only the UM-lignin containing composite obtained a V-0 rating in the UL94 test. A reduction of the peak heat release rate by 64–75% was achieved for both composites with the UM-lignin formulation having the better performance in terms of cone calorimeter tests. Scanning electron microscopy (SEM) and X-ray photoelectron spectroscopy (XPS) confirm denser char morphology with more phosphorus-containing char for the APP/UM-lignin system.

Costes et al. carried out a simple phosphorus/ nitrogen modification on two different lignins (kraft and organosolv) and evaluated their flame retardant effect in PLA [161]. Upon incorporation of 20 wt% of lignin the pHRR decreases by 30% but only the chemically modified lignin reached a V-0 rating in the UL94 test. Nevertheless, gel permeation chromatography (GPC) results confirmed thermal degradation of PLA when adding lignin, no matter if modified or not. In order to limit the decrease in molecular weight induced by the addition of lignin, Costes et al. combined lignin with phytic acid [162]. The presence of the acid enabled better dispersion of lignin into the PLA matrix and reduced the thermal degradation caused by it. Likewise, lignin reduced the composite hygroscopy induced by the presence of phytic acid. Moreover, incorporation of the phytic acid-lignin combination leads to an improved fire behavior. With the addition of 10 wt% lignin and 10% phytate in PLA a reduction of the pHRR by 44% was achieved while obtaining a V-2 classification in the vertical UL94 test.

The incorporation of several metallic phytates (Na-, Al-, Fe-, La-Phyt) with different oxidation states of the metal centers in PLA was studied by Costes et al. as well [163]. Aluminum phytate was proven to be the best compound compared to the other salts regarding their flame retardant properties but also being the most efficient initiator for PLA degradation during melt processing. To limit reduction of the molecular weight Al-Phyt was combined with Na-Phyt, which triggered a reduced degradation during melt blending in PLA. The Na-/ Al-Phyt (15 wt%/5 wt%) combination reduced the pHHR by 45% and obtained a V-2 rating in the UL94 test. Inductively coupled plasma (ICP) analysis of the char formed during the combustion showed that the used metallic phytate salts act in the condensed phase.

The addition of a novel phytate salt based on tannic acid and polyethyleneimine (Phyt/PEI-TA) in PLA was investigated by Laoutid et al. [164]. Upon incorporation of 20 wt% (Phyt/PEI-TA) the pHRR was decreased by 37%. Due to IPC analysis, it was found that the phosphorus remained mainly in the condensed phase confirming the condensed phase action of Phyt/PEI-TA.

The flame retardant effect of Al-Phyt in PLA was evaluated in combination with phosphorylated cellulose (P-cellulose) [165]. The Al-Phyt/ P-cellulose (5 wt%/10 wt%) composite allowed a reduction of the pHRR by 38% and reached a V-2 classification in the UL94 test due to the formation of a stable insulating char layer.

Fox et al. used polyhedral oligomeric silsesquioxane (POSS)–modified nanofibrillated cellulose (PNFC) as a carbon source in intumescing flame retardant formulations in PLA [166,167]. The bulky POSS groups inhibited the general problem of degradation of PLA when using APP due to a reduced phosphate accessibility to the polymer backbone. In the cone calorimeter test the PLA/APP/PNFC (11.25 wt%/3.75 wt%) composite showed a reduced pHRR by 46%, being slightly better than the comparable composite containing APP/PER (11.25 wt%/3.75 wt%). Both composites obtained a V-0 rating in the UL94 burning test.

Vahabi et al. investigated the application of a biobased flame retardant containing hydroxyapatite, perylene and lignocellulose (LHP) in PLA. When incorporated in combination with APP at a 1:3 ratio (LHP: APP) the pHRR was decreased by 32% in the cone calorimeter test due to an enhanced char formation [168].

Chen et al. evaluated the effect of chitosan (CS) on the flammability of PLA/APP composites [169]. PLA containing a combination of 5 wt% APP with 2 wt% CS led to an LOI of 33%, passed the UL94 V-0 rating and decreased the pHRR by 17%. Fourier transform infrared spectroscopy (FTIR) and TGA results indicate that CS acts as carbon agent due to its high content of carbon while characterization of the char residue by SEM confirmed the formation of a dense, homogeneous and continuous char layer.

Feng et al. investigated the addition of *β*-cyclodextrin (CD) as a carbon source in an intumescent flame retardant system consisting of APP and melamine (MA) [170]. When introduced in PLA, the corresponding composites had an increased LOI value of 34% and reached a V-0 classification in the UL94 test. Wang et al. prepared an inclusion complex between CD and poly(propylene) glycol (PPG) and added it in combination with APP and MA into PLA via melt blending [171]. This system exhibited more effective carbonization compared to the composite containing free CD. A LOI value of 34% and V-0 rating were obtained while the pHRR was decreased by 73%.

Wang et al. prepared PLA/APP/starch composites and evaluated the flame retardant properties by LOI, UL94 test and MCC [172]. Upon incorporation of 20 wt% APP and 10 wt% starch a LOI value of 41% and a reduction of the pHRR by 76% was reached. Moreover, a V-0 rating without any melt dripping behavior was obtained in the UL94 test.

Zhang et al. incorporated casein, a phosphoprotein derived from milk with a high phosphorus and nitrogen content, into PLA via melt compounding [173]. The results showed that the introduction of 20 wt% casein led to composites with an increased LOI value of 32% and a decreased pHRR by 18%. In the UL94 test a V-0 rating was achieved. Mechanism analysis by FTIR and SEM indicated that casein took effect in both the gas phase by releasing nonflammable gases and the condensed phase by the formation of a char layer.

### 3.2. Poly (Ethylene Therephthalate)

Poly(ethylene terephthalate) as engineering plastic is used in many applications like packaging or construction materials due to its lightweight properties and stability. Triphenylphosphate (TPP) has toxicological effects but it can be incorporated in the cavities of *β*-cyclodextrin (CD) forming an inclusion complex (IC) to decrease these effects while the flame retardant efficacy is maintained in hot pressed PET films. Different formulations were made containing CD, TPP or the inclusion complex in PET. The film thickness was about 0.3 mm and flame retardancy tests showed that PET containing only CD burned completely while PET containing IC or TPP had self-extinguish properties with shorter burining times for the PET-IC film [174].

In a layer-by-layer process PET foams are coated with DNA (Scheme 6) and compared with a coating containing APP. The flame retardant coating consisting of DNA was homogeneously allocated on the surface and able to reduce the heat release rate peak by 7% whereas the performance of APP reduced the heat release rate peak by 25% when 4 quad-layers are applied on the foams. The coating containing DNA was not able to prevent the melt dripping in contrast to the APP containing coating. Both flame retardants had no significant effect on the time to ignition [105]. Alongi et al. applied a homogenous DNA coating on bulk PET (thickness 3 mm) by a hot compression molding process at 120 °C resulting in a total amount of 10 wt% DNA in the polymer sample. A reduction of peak heat release rate by 42% could be achieved under 35 kW/m^−2^ [107].

Cotton fabrics were treated with an aqueous DNA solution by an impregnation process by Alongi et al. and showed no ignition during cone calorimetry tests with an irradiative heat flux of 35 kW/m^−2^ and an increase of LOI by 10% compared to untreated cotton fabrics [111]. Further investigations showed that a concentration of 19 wt% DNA leads to a fire resistance when the cotton fabrics are treated with an irradiation of 35 kW/m^2^ in the cone calorimeter leading to a pyrolysis instead of burning [110].

Alongi et al. demonstrated that DNA is suitable as a coating for EVA, PP, PET, PA6 and ABS bulk samples. The fire behavior was evaluated by using cone calorimetry with an irradiation of 35 and 50 kW/m^−2^. 10 wt% DNA content in the whole sample has emerged as an optimum concentration for 3 mm thick samples. For all samples the time to ignition was increased and the peak heat release rate was decreased significantly for an irradiation of 35 and 50 kW/m^−2^. Under 35 kW/m^−2^ the time to ignition could be increased by 342% (PET + 10 wt% DNA) up to 1637% (PA6 + 10 wt% DNA) and the peak heat release rate by 42% (PET + 10 wt% DNA) down to 57% (ABS + 10 wt% DNA) [107]. A layer-by-layer coating of DNA for PET foams was compared to a corresponding coating containing APP (ammonium polyphosphate) by Alongi et al. [105].

The fire and the anti-dripping behavior of PET fabrics could be improved by a deposition of chitosan and ammonium polyphosphate on the PET fabrics by using a layer-by-layer assembly technique. A 20 bilayer assembly on PET fabrics result in an increase of the LOI by 24% and no melt dripping during the vertical burning test [175].

A casein treated PET fabric results in an increased residue of 9% after a heat flux of 35 kW m^−2^ and a reduction in peak heat release rate of 2.7% but cannot prevent the melt dripping behavior. The time to ignition is decreased by 112 s for neat PET fabric to 62 s for casein treated PET fabrics. The horizontal flame spread test revealed a reduction of the burning rate by 67% and an increased residue of 34% [112,113].

Phosphorous-containing PER compounds were also tested for PET and PBT [104,176] DOPO-PEPA (Scheme 7) was used in PET fibers with a total phosphorus content of 0.8 wt% reaching a UL94 V-0 classification [176]. 5 wt% DOPO-PEA in PET resulted in a LOI value of 35.2% (neat PET: 24.6%) with no melt dripping behavior. The same amount of DOPO-PEPA in PBT led to a less intense melt dripping behavior an increase of the LOI value from 24.4% to 28.5% [104].

DOPO-IA (Scheme 5) can be synthesized via Michael addition type reaction of DOPO and itaconic acid and was developed by Toyo Boeski in 1975 [177]. A low molecular weight polymer consisting of DOPO itaconate and ethylene glycol was used for PBT, PA, TPE and PET [178]. Due to the acid groups of DOPO-IA it can be esterified resulting in different phosphorus-containing polyesters [101] or incorporated in the PET backbone via reactive extrusion (PET-P) [179]. PET-P has a total phosphorus content of 0.6 wt% and leads to an increase in the pyrolysis char yield in nitrogen atmosphere of 21% in comparison to neat PET with 17.9% [179].

25 wt% of GL-3DOPO (2.69% phosphorus content, Scheme 6) in PET resulted in 17.3% char residue at 750 °C compared to 7.9% for neat PET in nitrogen atmosphere. 25 wt% of GL-3DOPO in PET is suitable for a V-0 classification in the UL94 vertical burning test without burning drips. MCC tests showed a decrease of the multipeak heat release capacity (MHRC) of 481 Jg^−1^/K^−1^ in comparison to neat PET with 568 Jg^−1^/K^−1^. The LOI was increased from 22.8 to 35.4% (25 wt% GL-3DOPO in PET) for PET and also increased from 20.2 to 27.9% (25 wt% GL-3DOPO in PBT) for PBT. A total phosphorus content of 0.5% can achieve a V-2 classification in UL94 vertical burning tests whereas total phosphorus contents from 0.8% to 2.5% result in a V-0 classification for GL-3DOPO in PET. For PBT a total phosphorus content of 1.5% was able to achieve a V-0 classification and from 2.0% up to 2.5% a V-0 classification was obtained [103].

### 3.3. Poly (Butylene Succinate)

Polybutylene succinate (PBS) can be obtained by polycondensation of 1,4-butandiol and succinic acid, resulting in a biodegradable polymer. It is used for medical articles [180], food packaging [181], cosmetic bottles [182], etc. Hu et al. investigated flame retardants containing isosorbide modified with sulfur, phosphorus and silicon chlorides resulting in molecular or polymer flame retardants for polybutylene succinate. The phosphorus containing flame retardants are composed of phosphonates (DDI), phosphates (PPPAI, DPPI, PPPI) and phosphinates (DPPII) (Scheme 4) and were tested in PBS at a loading of 15%. Isosorbide modified with phosphorus showed better performance in terms of fire behavior than the flame retardants modified with silicon or sulfur. Regarding UL94 15 wt% of the phosphorus flame retardants respectively reached V-0 classification with non-burning dripping. 15 wt% of the phosphinate (DPPII) and the phosphonate (DDI) flame retardant in PBS in each case lead to an increased time to ignition of 38 s (DPPI) and 95 s (DDI). The phosphate flame retardants (PPPI, PPPAI, DPPII) promote the formation of char but the peak heat release rates are significantly increased up to 107% [97].

Ferry et al. grafted phosphoros compounds onto lignin for PBS and compared it with unmodified lignin whose flame retardant properties are known in the literature [51,52]. Ferry et al. compared the two neat lignins also by using cone calorimetry and reported different pHRR and TTI (time to ignition) for the alkali (102 kW/m^−2^ and 189 s) and organosolv (111 kW/m^−2^ and 25 s) lignin. Unmodified lignin in PBS leads to a decreased time to ignition and heat release rate with charring behavior. In comparison, lignin modified with phosphorus-containing homo- and copolymers in PBS was able to improve the fire behavior due to the formation of a more stable char. In cone calorimetry tests the THR was slightly decreased from 22.1 kJ/g^−1^ (neat PBS) down to 19.0 kJ/g^−1^ (PBS + 20% Lig Alk) and the pHRR was halfed. Phosphorus-modified lignin was able to increase the pHRR reduction from 562 kW/m^−2^ (neat PBS) down to 270 kW/m^−2^ (PBS + 20% Lig AlkcopoP) [53].

## 4. Biobased Flame Retardants in Epoxy Resins

The different epoxy monomers shown in Section 2.3.1 were designed for different applications. Therefore, the different resins cannot be compared to each other, but to a reference system. Ménard et al. [62] incorporated P2EP1P to the systems DGEBA/IPDA, P3EP/IPDA, P3EP/DA10, P3EP/DIFFA (IPDA = isophorone diamine, P3EP = triglycidyl phloroglucinol, DA10 = decane-1,10-diamine, DIFFA = difurfurylamine) yielding 1% or 3% phosphorus content in the matrix. Systems containing P2EP1P lead to lower pHRR and THR in pyrolysis combustion flow calorimetry (PCFC) because of a higher char yield (detected in TGA at 700 °C) and EHC (effective heat of combustion). It was shown, that the residue at 700 °C is over 30% for 3% P in DGEBA/IPDA and only 9% without the incorporated flame retardant. The pHRR is analogically lowered by 50%. This effect is also observable for systems with lower P-loadings, but less distinctive. On the other hand, the glass transition temperature of the matrix is lowered by the incorporation of P2EP1P significantly up to 50 °C.

Ma et al. [45] investigated the system consisting of DGEBA/EADI/MHHPA (MHHPA = methyl hexahydrophthalic anhydride) in different compositions between 100/0/68 and 52/48/62 as well as a system without DGEBA 0/100/56. This leads to phosphorus contents between 0% and 4.4%. The sample containing 4.4% phosphorus without DGEBA achieves a V-0 rating in UL94 vertical flammability tests. The glass transition temperature of this sample is 102 °C and therefore 8 °C lower than the glass transition temperature of the sample without EADI. Ma et al. deduce this observation from a lower crosslinking density because of steric hindrance in EADI and MHHPA. The change in flexural strength is within the error range for the different samples.

Neat DOPO-III or a combination with 10-undecenoyl triglyceride (UDTGE) or 3,4,5-tris (10-undecenoyloxy) benzoate (UDBME) was cured with 4,4′-diaminodiphenylmethane (DDM) or bis(*m*-aminophenyl) methylphosphine oxide (BAMPO) by Lligadas et al. [60]. Resulting thermoset materials show increased LOI values for phosphorus containing samples and glass transition temperatures at 108 °C with DDM and 95 °C with BAMPO.

Eugenol and cardanol based flame retardant epoxy monomers were tested after curing with MXDA (*meta*-xylenediamine) by Calliol et al. [41] with DGEBA as reference epoxy resin. The phosphorus content in the matrix depends on the monomer structure and varies between 2% and 6%. The glass transition temperatures for eugenol based resins are about 100 °C lower than for the DGEBA-based system. PCFC tests show that the pHRR is lowered by 25% for phosphorus containing samples, whereas the residue is increased by up to 55%. These results are similar to results obtained by Miao et al. [183] curing TEEP with DDM.

Flame retardant curing agents were tested by partially replacing non-biobased curing agents in epoxy resin systems. Dai et al. [135] cured DGEBA with DDM and DOPO-PHM. A 50/50 mixture of these curing agents and a resulting phosphorus content of 1.7% in the matrix leads to an increase in LOI up to 33%. The char yield at 700 °C under nitrogen atmosphere is increased by 6%. A single glass transition temperature around 100 °C indicates a homogenous material.

The bisphenolic structure Ar-DOPO-N synthesized by Liu et al. [36] was tested in reference to *o*-cresol formaldehyde novolac epoxy resin (CNE) cured with phenol formaldehyde novolac (PFN) and melamine-modified novolac (MPFN) in different ratios. The resulting glass transition temperature is decreased by 20 °C for samples with 30 wt% flame retardant curing agent in the matrix. The LOI is simultaneously raised from 21% to 33%. The residue at 700 °C is increased from 30% for the sample without phosphorus to over 50% for the sample with 30 wt% AR-DOPO-N incorporated. A P-N synergistic effect was shown for the melamine-containing system.

Hydroxyl PES was designed as a flame retardant curing agent for epoxy films by Lin et al. [136] and tested in DGEBA, CNE and HP-7200. Resulting systems are transparent, tough, achieve V-0 in UL94 vertical tests setups and have a high glass transition temperature of about 260 °C. An application for LED encapsulation is suggested.

The myrcene-based MMDOPO prepared by Yang et al. [57] was cured with DGEBA and DMP as accelerator. While the resulting material showed an increased LOI and a moderate glass transition temperature of 100 °C, it was brittle at the same time. Hence, it was combined with a phosphate containing castor oil derived curing agent combining balanced strength and elongation at break.

Polyesters by Wang et al. [69] were cured with epoxidized soybean oil. Only samples with a higher phosphorus content than 2% achieved a V-2 classification in UL94 tests. Flame retardant efficiency was shown by microcombustion calorimetry (MCC). The peak heat release rate was decreased by 50% and the total heat release was decreased by 7%. For a possible application in PSA tapes, the samples were tested additionally after procedure GB/T 15903-1995 [184], achieving an enhanced result (level 1) because of an observed self-extinguishment. Reference samples consisting of polyesters without incorporated phosphorus groups burned completely.

TECP containing DOPO-groups were incorporated to DGEBA/DDM by Hu et al. [18]. A flame retardant content of 30 wt% in the matrix shows an increased LOI by about 30%. The char residue at 800 °C in nitrogen atmosphere is 8% and the pHRR was decreased by 50% in cone calorimeter measurements compared to DGEBA/DDM samples. Mechanical tests showed increased impact strength.

The nano-filler obtained from phytic acid was dispersed in DGEBA/m-DPA (*meta*-phenylenediamine) and achieves a V-0 classification in UL94 tests and a LOI of about 30% for samples with 20 wt% or 30 wt% PAMA incorporated to the matrix. The pHRR was decreased by 45% in reference to a system cured with *N*-aminoethylpiperazine, while the residue after cone calorimeter tests is increased from 3% to 35%.

The determined inclusion complex showed a delay in decomposition referring to the flame retardant molecule without the inclusion complex. Zhao et al. [145] detected an increased residue at 600 °C and a decreased pHRR by 36% for 2 wt% incorporated inclusion complex. They propose a flame retardant mode of action in the gas and condensed phase.

PER based flame retardant PFR was tested in DGEBA/DDM [138]. At 15 wt% PFR content the system achieved V-0 in vertical UL94 test and the pHRR was lowered by 60% in MCC. The glass transition temperature is higher for samples containing PFR. Because of the high loading tensile strength is lowered to 17 MPa, whereas elongation at break was higher by 50–70%.

DOPO-PEPA was tested by Zhang et al. [106] in DGEBA/DDM. At 5.7wt% DOPO-PEPA content the system achieves a V-0 classification and a LOI of about 30%. Also the pHRR was decreased by 32% to 873 kW/m^−2^. A reference system with 9.1 wt% DOPO achieved similar results in cone calorimeter and LOI tests. But it achieved no rating in UL94 testing.

## 5. Outlook

Within a few years, many biobased flame retardant systems were developed, but an industrial scale up will not be feasible so fast. A successful flame retardant is determined by the flame retardant efficiency or potential flame retardant formulations. We showed that there are many different systems available, which can be improved through ongoing research. Promising flame retardant structures for epoxy resins are realized with biobased epoxides or curing agents. Such compounds from petrochemical sources are bio-accumulative and have negative effects on health of human and environment. On the other hand, biobased molecules are not per se healthy. Chemically modified they are equal to their petrochemical equivalents und thus have the same health safety issues. Therefore, toxicology of all substances still must be determined. As we can see from BPA, some structures are available from renewable sources, but from an economical point of view, they are not profitable, because the petrochemical alternative is less expensive to produce. In terms of rising crude oil prices, the research on alternative production chains is important for the future. Some production chains, which are too expensive nowadays, will be viable in future. The cost of an industrial scale up depends on required technologies and raw materials. Some of the presented materials are based on special organisms and others on byproducts of industrial processes like wine or wood production. These byproducts which are available in large scales are more prone to be considered raw materials for new biobased flame retardant systems.

The increasing market and interest in biobased polymers and biobased additives like flame retardants is a positive trend but during the replacement process recycling and the whole production process of biobased polymers including the extraction of raw materials are essential aspects that have to be considered in the evaluation of the life cycle assessment (LCA). The recycling process of plastics which contain additives like flame retardants (biobased or non-biobased) is more difficult than the recycling of neat polymers, because the effects of the additives can be reduced or the polymer characteristics can be influenced throughout the recycling loop. Research in this area is still needed and an important challenge in the future. The limited access to fossil-based resources encourages the growth of the biobased polymers and biobased platform chemicals due to identical properties of end products. There is a change in the plastic industry because the biodegradable plastics are no longer attractive for niche applications. Certain production processes for raw materials and chemicals made of renewable sources require water and land use which are no longer available for food production generating a competition of land use.

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
