# Peer review of "Phosphorus-Containing Flame Retardants from Biobased Chemicals and Their Application in Polyesters and Epoxy Resins"

_molecules, 2019, doi:10.3390/molecules24203746_

Round 1

Reviewer 1 Report

This manuscript summarizes the synthesis methods of biobased flame retardants and their applications in polyesters and epoxy resins. The choice of the discussed literature is excellent, some re-structuring (see the comments below) would highly increase the readability of the text.

Comments and suggestions:

chapter 1

You could refer to relevant IUPAC definitions:

Biobased polymer derived from the biomass or issued from monomers derived from the biomass and which, at some stage in its processing into finished products, can be shaped by flow.

Note 1: Bioplastic is generally used as the opposite of polymer derived from fossil resources.

Note 2: Bioplastic is misleading because it suggests that any polymer derived from the biomass is environmentally friendly.

Note 3: The use of the term "bioplastic" is discouraged. Use the expression "biobased polymer".

Note 4: A biobased polymer similar to a petrobased one does not imply any superiority with respect to the environment unless the comparison of respective life cycle assessments is favourable.

Terminology for biorelated polymers and applications (IUPAC Recommendations 2012)

Pure Appl. Chem., Vol. 84, No. 2, pp. 377–410, 2012

Please check the references /crossreferences, references to Figures, many of them were lost probably during the conversion to pdf (Error! Reference source not found.).

Please check the numbering of the Figures throughout the manuscript, as it is not consistent.

line 54

The land use for biopolymers could be added from your reference [8], page 46.

line 73

“foams in construction materials like wind turbine blades”

Are the wind turbines really made from polyester foams? Not from glass fibre reinforced polyester composites?

If you mean sandwich composite structures, their cores are rather made from other rigid foams as PUR.

life 78

life cycle assessments (LFA) – please check the abbreviation

line 89

“Another disadvantage of DGEBA-based epoxy resins is their inflammability [22].” – please check this sentence

line 99

Please add some introduction to this chapter, e.g. which starting materials, which phosphorylation methods are going to be addressed.

Also mention the issue, that although the starting materials are biobased most of the applied phosphorylating agents are not. Therefore the biobased ratio of the final product is necessary to assess the real impact of these FRs.

line 112

Abbreviations DDI, DPPI, DPPII and IDEA should be explained.

Please add reference(s) to Figure 3.

Figure 5

Direct reaction of PER with DOPO is also described in the literature:

https://pp.bme.hu/ch/article/view/2175

The numbering of figures restarts in chapter 2.2.

Either add the number of the chapter to the figure, or renumber the figures throughout the manuscript.

line 185

Itaconic acid is also suitable to crosslink epoxy resins directly.

You can also check: Green Chem., 2013, 15, 245 DOI: 10.1039/c2gc36715g

The structure of the manuscript could be made more consistent.

This is the current structure:

Synthesis of phosphorus-containing biobased flame retardants

2.1. Phosphorylation of alcohols -

2.2. Phosphorylation on double bonds

2.3. Phosphorus-containing biobased flame retardants for coatings

2.5. Phosphorus-containing biobased flame retardants for epoxy resin systems

2.5.1. Phosphorus-containing, reactive flame retardant epoxy monomers

2.5.2. Phosphorus-containing, reactive flame retardant curing agents

2.5.3. Phosphorus-containing, non-reactive flame retardants

Biobased flame retardants in polyesters

3.1. Poly (lactic acid)

3.2. Poly (ethylene therephthalate)

3.3. Poly (butylene succinate)

Biobased flame retardants in epoxy resins

4.1. Flame retardant properties

Comments, suggestions:

Chapter 2.4 is missing.

Chapters 2.1 and 2.2 group the synthesis methods on the basis of the starting material, while chapter 2.3 on the basis of application, chapter 2.5 on the polymer, where the FRs are going to be used.

Please make the grouping of the synthesis methods more consistent.

In chapter 4. Biobased flame retardants in epoxy resins there is only one subchapter 4.1. Flame retardant properties, and the whole chapter 4 is about the FR results, consequently the title subchapter 4.1 is unnecessary.

line 268

“Cardanol or eugenol undergoes a condensation of HCl with phosphoryl chloride or a derivative”

Please check this sentence, maybe elimination of HCl is more appropriate.

line 283

“The reaction of cardanol and phosphoryl trichloride with followed epoxidation to the flame retardant…”

Please check this sentence, maybe this version is easier to understand:

The reaction of cardanol and phosphoryl trichloride followed by epoxidation led to flame retardant…

Figure 15 should be after Figure 5 if we follow the order of the relevant text.

Please correct the style of line 311-316. Reference to Figure 17 is missing.

In chapter 2.5 only the synthesis methods are summarized, but the related FR results are completely missing (except the line 349). In the previous subchapters of chapter 2 FR results were also included, which makes this chapter inconsistent.

The suggestion of the reviewer is either to discuss the FR results in all cases directly after the description of the synthesis method (this helps the reader a lot to understand the results, as the chemical structure are embedded there in the manuscript), or discuss all the FR results in chapters 3 and 4.

Line 329

“The reactivity of oxirane groups can be used adding different functional groups.”

Please replace to:

The reactivity of oxirane groups can be used to add different functional groups.

line 500

Please check this sentence:

“The flame retardant coating consisting of DNA was homogeneously and able to…”

line 548

“Ferry et al. grafted phosphorous compounds onto lignin for PBS…”

The spelling "phosphorous" should be reserved for the trivalent state of phosphorus, as in phosphorous acid, H3PO3.

line 559

„Therefore, the different resins cannot be compared to each other, but to a reference system.”

Even though the reference systems are different, they could be summarized in a form of a table (e.g. including columns as reference matrix, name and abbreviation of the flame retardant(s), P content in mass%, most important FR results (LOI, UL-94, peak of heat release rate, if available), any other comments, references).

Author Response

Dear Dr. Sonnier, Pr. Ferry, Dr. Vahabi,

Our manuscript was thoroughly revised according to the suggestions/comments of the reviews.

In the following we describe how the manuscript has been altered and supplemented:

Abbreviations were explained when used for the first time

Comments from Reviewer #1:

 The definition of biobased polymers is referenced by an IUPAC definition (line 27-31). It was clarified that biobased polymers not necessarily are environmentally friendly (line 28-30). The term “bioplastic” was replaced by the term “biobased polymer” in the whole manuscript. The terminology for biorelated polymers and applications by IUPAC was used and it was clarified, that a favorable life cycle assessment in terms of environmental aspects is required for a superiority of the biobased polymer over its petrochemical derivative in line 88-90.

 The references, crossreferences, references to figures and the numbering of the figures was checked and corrected.

The land use for biobased polymers was added from reference [8] (lines 58-61).

 For the production of wind turbine blades, common materials for the foam core are: PET, PVC and balsa wood. For example:

ArmaForm (https://local.armacell.com/en/armaform-pet-foam-cores/markets/wind/)

BASF Kerdyn (https://www.materialstoday.com/composite-parts/news/basf-introduces-pet-foam-for-wind-turbine-blade/)

Mishnaevsky, Leon, et al. "Materials for wind turbine blades: an overview." Materials 10.11 (2017): 1285.

Fathi, Amir, Jan-Hendrik Keller, and Volker Altstaedt. "Full-field shear analyses of sandwich core materials using Digital Image Correlation (DIC)." Composites Part B: Engineering 70 (2015): 156-166. Additional references were added to this paragraph.

 The abbreviation of life cycle assessment was checked and corrected (line 88).

 The word “inflammability” was replaced by “flammability” to avoid confusion; also there is no difference in the meaning of “inflammability” and “flammability” (line101).

 An introduction to chapter 2 was added, addressing the phosphorylating methods and starting materials. It was mentioned that most of the phosphorylating agents are not biobased but the biobased ratio can be enhanced by phosphorylating biobased starting materials (lines 114-121).

 The abbreviations of DDI, DPPI, DPPII and IDEA were added (lines 169-171).

 Suggested references were added to the text (line 269).

 The numbering of all figures was checked and corrected.

The chapters are rearranged. This is the new structure:

Introduction Synthesis of phosphorus-containing biobased flame retardants

2.1. Biobased flame retardants for Poly (lactic acid)

2.2. Biobased flame retardants for Poly (ethylene therephthalate) and Poly (butylene succinate)

2.3. Biobased flame retardants for epoxy resin systems

2.3.1. Reactive flame retardant epoxy monomers

2.3.2. Reactive flame retardant curing agents

2.3.3. Non-reactive flame retardants

Biobased flame retardants in polyesters

3.1. Poly (lactic acid)

3.2. Poly (ethylene therephthalate)

3.3. Poly (butylene succinate)

Biobased flame retardants in epoxy resins Outlook

 The sentences were rephrased as proposed (lines 274-275, 289-291, 346, 526-527, 595-598).

FR results were moved to chapter 3 and 4 where all FR results were discussed.

Even though a summary in form of a table would be a clear presentation of the flame retardant results, it would indicate a possible comparison between different resin systems although different flame retardant mode of actions are required depending on the application and resin system. Additionally the phosphorus contents and the fire tests (LOI, UL94 and cone calorimetry) are not consistent enough for a comparison so that the summary in form of a table would be incomplete.

We hope that you comply with the revision of our manuscript and publish it in near future.

Sincerely Manfred Döring

Reviewer 2 Report

This is a very interesting and valuable review. However, some corrections (mainly editorial) are necessary:

1. on page 3, in Table 1 - "Acidic acid hydrolysis ...",

2. on p. 4, in line 109 - an abbreviation of "DOPO" has not been explained,

3. on pages 3 and 4, in Table 1 - abbreviations of many polymers (PBAT, PBS, PBT, PET, PEF, PBF, PLA, PTT) should be explained,

4. on p. 4, in line 112 - the abbreviations of "DDI, DPPI, DPPII and IDEA" should be explained,

5. on p. 5 - a therm "TTI" (in line 138) and the abbreviation of "DMAP" (in Fig. 3) should be explained,

6. numbers of many Figures should corrected:

on. p. 8 - Fig. 1 for Fig. 7, on p. 10 - Fig. 2 for Fig. 11, on p. 11 - Fig. 2 for Fig. 11  and Fig. 3 for Fig. 13, on p. 12 - Fig. 4 for Fig. 14 and Fig. 5 (in lines 295, 298 and 302) for Fig. 15, on p. 13 - Fig. 5 for Fig. 16 and Fig. 6 for Fig. 18,

7. many other abbreviations of FRs presented in Figures or in a text of the manuscript, on pages 6-15 and elsewhere (e.g. APP), should be explained.

8 Which structure corresponds to "DOPO-HQ" (see line 321 and corrected Fig. 18 on p. 13) ?

9. A name "Hexamethylendiamine" in line 181 on p. 8 should be changed for "hexamethylene diisocyanate".

10. A sentence: "Error! Reference source not found." must be removed from whole manuscript. If necessary, please provide proper references !

11. What was an effect of use of DOPO-PEPA, as flame retardant in PET [93,94], PBT [93] and epoxy resins [93,95], on flammability of above polymers ?

Author Response

Dear Dr. Sonnier, Pr. Ferry, Dr. Vahabi,

Our manuscript was thoroughly revised according to the suggestions/comments of the reviews.

In the following I describe how the manuscript has been altered and supplemented:

Abbreviations were explained when used for the first time

Comments from Reviewer #2:

Table 1 "Acidic acid hydrolysis ..." was corrected. The abbreviation of DOPO was added on page 4. The abbreviations of PBAT, PBS, PBT, PET, PEF, PBF, PBF, PLA and PTT were added in table 1. The abbreviations of DDI, DPPI, DPPII and IDEA were explained.

5.The abbreviations of TTI and DMAP were added.

The numbering of the figures were checked and corrected. The abbreviations of APP and other used FR were added in the manuscript. The name DOPO-HQ was added to the figures.

9.”Hexamethylenediamine” was replaced by “hexamethylene diisocyante”.

All references were checked and corrected. The flame retardant effect of DOPO-PEPA in PET, PBT and epoxy resin was added.

We hope that you comply with the revision of our manuscript and publish it in near future.

Sincerely Manfred Döring